# Selecting and Tailoring Implementation Strategies to Improve Human Papillomavirus Vaccine Uptake in Zambia: A Nominal Group Technique Approach

**DOI:** 10.3390/vaccines12050542

**Published:** 2024-05-15

**Authors:** Mwansa Ketty Lubeya, Mulindi Mwanahamuntu, Carla J. Chibwesha, Moses Mukosha, Mary Kawonga

**Affiliations:** 1Department of Obstetrics and Gynaecology, School of Medicine, University of Zambia, Lusaka 10101, Zambia; mulindim@gmail.com; 2Women and Newborn Hospital, University Teaching Hospitals, Lusaka 10101, Zambia; 3School of Public Health, Faculty of Health Sciences, University of the Witwatersrand, Johannesburg 2017, South Africa; moses.mukosha@unza.zm (M.M.); mary.kawonga@wits.ac.za (M.K.); 4Clinical HIV Research Unit, Helen Joseph Hospital, Johannesburg 2193, South Africa; carla_chibwesha@med.unc.edu; 5Department of Pharmacy, School of Health Sciences, University of Zambia, Lusaka 10101, Zambia; 6Department of Community Health, Charlotte Maxeke Johannesburg Academic Hospital, Johannesburg 2193, South Africa

**Keywords:** feasibility, acceptability, barriers, cervical cancer elimination, Zambia, knowledge, myths and misconceptions, implementation strategies

## Abstract

The human papillomavirus (HPV) vaccine is effective in cervical cancer prevention. However, many barriers to uptake exist and strategies to overcome them are needed. Therefore, this study aimed to select and tailor implementation strategies to barriers identified by multiple stakeholders in Zambia. The study was conducted in Lusaka district between January and February 2023. Participants were purposively sampled from three stakeholder groups namely, adolescent girls, parents, and teachers and healthcare workers. With each of the stakeholders’ groups (10–13 participants per group), we used the nominal group technique to gain consensus to tailor feasible and acceptable implementation strategies for mitigating the identified contextual barriers. The identified barriers included low levels of knowledge and awareness about the HPV vaccine, being out of school, poor community sensitisation, lack of parental consent to vaccinate daughters, and myths and misinformation about the HPV vaccine. The lack of knowledge and awareness of the HPV vaccine was a common barrier across the three groups. Tailored strategies included conducting educational meetings and consensus-building meetings, using mass media, changing service sites, re-examining implementation, and involving patients/consumers and their relatives. Our study contributes to the available evidence on the process of selecting and tailoring implementation strategies to overcome contextual barriers. Policymakers should consider these tailored strategies to mitigate barriers and improve HPV vaccine uptake.

## 1. Introduction

Human papillomavirus (HPV) vaccination serves as a primary prevention strategy in reducing the global incidence of cervical cancer to less than 4/100,000 women, referred to as the elimination of cervical cancer [1]. In 2020, the World Health Organisation (WHO) issued a global call with three specific targets to be achieved by 2023 to make progress towards the elimination of cervical cancer as a public health problem [1]. The specific targets include the following: “90% of girls fully vaccinated with HPV vaccine by age 15 years, 70% of women are screened with high-performance test by 35 years of age and again by 45 years of age, 90% of women with precancer treated and 90% of women with invasive cancer managed” [1]. Thus, the focus of this paper is on the first strategy.

The current global statistics for the HPV vaccine show that uptake of this important vaccine remains very low, especially in very vulnerable communities of adolescent girls [2,3,4]. The implementation of the HPV vaccination has been faster in high-income countries compared to low-income countries, with a global population coverage of 15% [2] owing to multiple challenges [5,6,7,8,9,10,11,12]. Systematically reviewed data from 2006 to 2014 showed that in more developed regions, 33.6% of females aged 10–20 years received the full course of vaccine compared with only 2.7% in less developed regions [13].

Zambia has not been spared from low HPV vaccine coverage due to various barriers [5,14,15,16,17] despite having one of the highest cervical cancer-related morbidity and mortality rates in the world [18,19,20]. The high incidence is further compounded by the high prevalence of HIV among women of reproductive age, late presentation to health facilities, and delayed diagnosis [21,22,23,24,25] despite the country having a robust cervical cancer screening program since 2006 [26,27,28]. Since 2019, the HPV vaccine in Zambia has been offered for free to a single cohort of 14 year old girls in health facilities and outreach facilities such as schools, where most eligible girls are found. A detailed description of the implementation of the HPV vaccination in Zambia is published separately elsewhere [5].

To achieve the goal of high HPV vaccination adoption and successfully contribute to the global call for cervical cancer elimination [1], the immunisation agenda of 2030 [29], and the sustainable development goals, evidence-based implementation strategies are important to mitigate contextual barriers to HPV vaccine implementation. Implementation strategies are actions for promoting the use of research evidence and evidence-based interventions (such as the HPV vaccine) in real-world practice [30]. For instance, physician training, assessment, and feedback are specific strategies shown to be effective in increasing HPV vaccine reach and uptake. Specific actions may include strategies targeting providers as well as educating patients and parents. Implementation strategies should be tailored to a particular context for them to be relevant as strategies which have been used in a particular set up may not necessarily apply to another context [31,32]. 

There are several implementation strategies which have been used for implementing HPV vaccination programs in SSA [33,34]. However, the selection and tailoring of strategies to contextual implementation barriers are underdeveloped [31]. Tailoring in this study was defined as a selection of implementation strategies to address contextual barriers which may influence the uptake of the HPV vaccine. In Zambia, there has been limited effort in engaging key stakeholders in the process of identifying implementation barriers and selecting and tailoring strategies for overcoming these. This paper addresses this gap.

We used the nominal group technique (NGT) [35] to identify stakeholder-perceived-specific barriers to HPV vaccination and tailor feasible and acceptable strategies to mitigate these barriers through consensus building. The NGT is proven to be beneficial in program planning including in healthcare [36,37]. Participants in a group come up with solutions which represent the groups’ preferences [38]. Therefore, in this study, we used the NGT to obtain stakeholder consensus on the most important barriers to HPV vaccination implementation and uptake [5,39]. Stakeholders further tailored implementation strategies to mitigate the identified barriers while prioritising strategies based on their perceived feasibility and acceptability.

## 2. Materials and Methods

### 2.1. Study Setting

The study was conducted within the six administrative subdistricts of Lusaka District, Zambia, between January and February 2023. This setting was selected as it was one of the regions where the HPV vaccination demonstration project was conducted between 2013 and 2017 and hence has some experience with the HPV vaccine. Healthcare facilities located within these subdistricts have the mandate to offer the HPV vaccine to eligible adolescent girls. The study setting is detailed elsewhere in a study conducted with this set up [5,39].

### 2.2. Study Design

We used the NGT for consensus building among the different stakeholders [35]. The nominal group technique is a structured method for group brainstorming that encourages contributions from everyone and facilitates quick agreement on the relative importance of issues, problems, or solutions. The process prevents the domination of the discussion by a single person as may happen in focus group discussions. The NGT typically includes 4 steps as listed below:Silent generation of ideas in writing (independently).Recording of ideas (round robin-no discussion at this point).Discussion of listed ideas (carried out for clarity of all ideas).Voting to enable ranking of priority ideas.

### 2.3. Participant Selection

Adolescent girls were sampled from schools and the communities in the study setting where the broader formative research was conducted [39]. Parents were sampled from the communities and social settings, while healthcare workers and teachers were sampled from health facilities and schools, respectively. The sampling procedure is detailed in the formative research we conducted in this same setting [39]. Participants were sampled purposively to select participants likely to give the most information about barriers to HPV vaccination in their context and to select and tailor implementation strategies based on their feasibility and acceptability. Some of the participants had taken part in the formative research.

### 2.4. Data Collection and Analysis

The data collection process involved three facilitators. One facilitator took field notes, whilst the other presented the questions and the third one managed the group. Stakeholder-specific homogenous groups were invited to the meeting on different dates. The meetings took place either in community halls or school halls. About 10–13 participants were selected per group to allow for an organised discussion as well as diversity within the participants [36]. 

The data collection sessions were divided into two parts. Part one included the identification and voting for the most pertinent barriers as perceived by each particular stakeholder group in addition to those identified in the formative research, whose results are published elsewhere [5]. During the introduction of the activity, barriers were defined as things that may hinder or affect their willingness to vaccinate, give consent to the vaccination, or receive the HPV vaccine depending on the specific stakeholder group. Part two included selecting and tailoring implementation strategies suggested by the participants and later coding these strategies with the expert recommendation for the implementation change (ERIC) taxonomy [40].

#### 2.4.1. Part 1: Selecting Barriers

Step 1: Generating ideas—identification of barriers

A stakeholder-specific question on barriers to HPV vaccination was written down on a piece of manilla paper stuck onto the wall and was read out to the participants (Table 1).

A total of 15 minutes were given for participants to write as many perceived barriers as possible on a piece of paper. The facilitators went round to ensure that the instructions were clear to the participants. The rationale was to see the consistency in the barriers elicited in the formative stage and to ascertain if additional barriers would come up.

Step 2: Recording of barriers

This step is also referred to as a round-robin recording of ideas. Each participant was asked to read out the barriers they had written down, and these were written on a mounted manilla paper by one of the facilitators. A discussion of the barriers was not permitted at this point. 

Step 3: Discussing identified barriers

Participants were asked to explain what they understood about the written barriers. Each barrier was discussed to ensure that there was clarity on its importance. The facilitators constantly asked participants if there were any barriers that needed to be clarified. The identified barriers were later summarised with consensus and written down on a mounted manilla paper. The barriers identified in the formative research were also considered by participants.

Step 4: Voting on important barriers

Participants were asked to copy down the summarised barriers and vote independently and silently, giving a maximum score to the most important barrier and a score of 1 to the least important barrier based on the total number of barriers. Each participant read out the scores, which were recorded on a score sheet and were later summed up. The three most important barriers (with the highest total score upon voting) were then written down to be used in the subsequent step for selecting and tailoring implementation strategies.

#### 2.4.2. Part Two: Tailoring of Strategies Based on Important Barriers

Like in part one, the NGT was again used to gain consensus on which implementation strategies would be ideal to mitigate each of the prioritised barriers. A stakeholder-specific question on strategies which can be used to overcome the identified barriers to improve HPV vaccination adoption by adolescent girls was written down on a piece of manilla paper and read out to the participants. 

The question was as follows: what strategies can be used to overcome barriers identified in the HPV vaccination program? Participants were asked to copy the top three barriers and write down as many strategies as possible which could be used to mitigate each one of the three barriers prioritised in the previous steps, separately. They were asked to write down barriers they perceived to be feasible and acceptable. This was conducted independently and silently in approximately 15 min. The identified implementation strategies were later coded using the ERIC taxonomy [40] for reproducibility purposes.

The selection and tailoring of the implementation strategies followed the four steps of the NGT as described in part one to match the barriers to implementation strategies based on the perceived feasibility and acceptability. All the interactions were written down on manilla paper and stuck around the room for reference. Additionally, the discussions were recorded and transcribed for an accurate reference of the discussion. The top three most important barriers were listed by the stakeholder groups, and each was tailored to three implementation strategies. The final selected implementation strategies were coded using the ERIC taxonomy to increase replicability. 

## 3. Results

The characteristics of the participants are presented in table form for each of the different stakeholder groups. 

### 3.1. Adolescent Girls

There were 13 girls who were purposively sampled to participate in this study. Of these, eight were in school and five were out of school. Five girls had received at least one dose of the HPV vaccine. The mean age was 15 years. (Table 2).

The identified barriers were as follows, with the vote count in brackets: the lack of knowledge and awareness about the HPV vaccine/program (78); the lack of school attendance (63); parents not consenting to vaccination (61); myths and misconceptions such as the HPV vaccine is an experiment, vaccination is an initiation into satanism, the HPV vaccine can bring cancer, someone can become lame after receiving the HPV vaccine, and misinformation about cervical cancer prevention (51); external influence by friends and families (49); COVID-19 pandemic—some adolescents thought the vaccine was for COVID-19 rather than HPV infection prevention (34); and fear of vaccine side effects (32). The top three barriers with the highest votes were identified and tailored to implementation strategies. The most feasible and acceptable strategies were voted for as shown in Table 3.

### 3.2. Parents Who Have Adolescent Girls

Thirteen parents were invited and participated in the NGT to select and tailor implementation strategies. The mean age was 46, and six were male (46%), seven had at least secondary education (54%), nine were employed (69%), 11 had school-going vaccine-eligible girls (85%), and five parents confirmed that their daughter had received at least one dose of the HPV vaccine (38%) (Table 4).

The identified barriers with participant scores for each barrier were as follows in order of importance and vote counts in brackets, the lack of knowledge and awareness about the HPV vaccine (72), inappropriate venues set up for the vaccination (45), myths and misconceptions (44), timing of the HPV vaccination (44), COVID-19 pandemic (35), and the lack of resources (33). The three most important barriers were then matched with implementation strategies voted based on feasibility and acceptability (see Table 5).

### 3.3. Teachers and Healthcare Workers

For the teachers and healthcare workers group, the total number of participants was 10. Six were teachers while four were nurses (Table 6).

The identified barriers in order of importance were as follows: myths and misinformation in the community (104), the poor sensitization of teachers and healthcare workers (83), the lack of information dissemination programs to the community (81), the lack of knowledge and sensitization among teachers and healthcare workers (70), the lack of materials on HPV vaccine (70), language barriers during community sensitization (69), the lack of communication between teachers and healthcare workers (68), the age restriction for eligible girls (68), inadequate human resources (49), teachers prohibiting pupils from getting vaccinated (41), private schools not allowing vaccination teams in their premises (39), and the COVID-19 pandemic (e.g., vaccine and restrictions) (38). Table 7 shows the selection process of the strategies.

## 4. Discussion

Implementation strategies based on the identified barriers were successfully selected and tailored by multiple stakeholders using the NGT. There was an overlap between the implementation strategies and barriers between the groups. Both parents and adolescent girls voted for a lack of knowledge about HPV and the HPV vaccine as the most important barriers. Myths and misinformation were among the top three most important barriers for the parents’ group and teacher/healthcare group.

The barriers identified by the stakeholders are summarised as follows: low levels of knowledge and awareness, being out of school, poor sensitisation, the lack of parental consent, and myths and misinformation. These types of barriers have been frequently reported in various regions [41,42,43,44,45,46,47,48]. Based on these barriers, the selected and tailored implementation strategies were to conduct educational meetings and consensus-building meetings, use mass media, change service sites, reexamine implementation strategies, involve patients/consumers and their families, and alter the incentives. 

Most of these tailored strategies are frequently used within SSA to increase HPV vaccinations and they are of high feasibility and importance, according to our recent scoping review [33]. In this study, all three stakeholder groups commonly suggested the use of educational strategies to mitigate barriers such as low awareness and knowledge about the HPV vaccine, the lack of parental consent, the poor dissemination of information about HPV vaccines, and myths and misinformation.

Educational strategies have been used in many setups [33,49,50] even though their value is debatable in some cases [51]. Some innovative educational strategies have, however, been found to be effective. For instance, Drokow et al. [52] used a video-based education intervention to increase the knowledge and uptake of cervical cancer screening and HPV vaccinations and the increase in both was statistically significant. Another study in an urban clinic in the USA used an educational video to educate parents of adolescent girls. Girls whose parents watched the educational video were three times more likely to be vaccinated at the end of a seven-month period [53].

In a quasi-experimental study in Malawi, a mini magazine was created for vaccine-eligible adolescent girls to provide educational information and create household conversations around the HPV vaccine. The study found a positive correlation between the consumption of the mini magazine and HPV vaccine uptake among the girls [54]. Additionally, that study involved the primary HPV vaccine recipients sharing information with their parents to help them buy into the HPV vaccination program, similar to an implementation strategy suggested by our stakeholders in this study to involve patients/consumers and their relatives.

The suggestion by various stakeholders to implement educational strategies could be partly explained by the recent introduction of the HPV vaccine at the national level (2019), sharing information with key stakeholders once a year during the vaccination campaign, with most information and education materials being available mainly in English. Therefore, considering that educational strategies are not always effective, culturally sensitive educational information should be packaged in ways which are effectives such as videos, comics, and magazines [54,55].

Further, to increase knowledge and awareness, the reach of adolescent girls and debunking myths and misinformation, the implementation strategy of the use of mass media was frequently selected. This strategy has also been widely used to increase HPV vaccinations [33,56,57,58,59]. For example, in Rwanda, announcements about the HPV vaccine were made via newspapers, magazines, and radio as part of the communication strategy [60]. Rwanda has one of the highest HPV vaccine uptakes in the world.

Additionally, a systematic review and meta-analysis on communication strategies used to increase HPV vaccine uptake among adolescents in SSA found that community meetings, information posters, flyers, television, radio, and newspapers were useful to increase HPV vaccine completion [61]. Therefore, the use of mass media in disseminating information plays a key role in reaching different stakeholders and increasing HPV vaccine uptake [48,62,63]. The other advantage for Zambia to implement such an approach is that mass media platforms use the different local languages to deliver information which can be useful to increase HPV vaccine uptake. 

The strategy about changing service sites was tailored to the barriers related to being out of school and inappropriate vaccination venues to make the HPV vaccine more accessible and ultimately increase uptake. In most SSA countries including Zambia, schools serve as the key primary sites for HPV vaccination [33,64,65]. School delivery has advantages such as reaching more eligible girls [66,67,68,69] and is more cost effective. However, this disadvantages out-of-school girls who are mainly found in hard-to-reach locations [70,71,72,73].

In most instances, despite out-of-school girls being more vulnerable, they are difficult to find during vaccinations [74,75]. Further, most school-girls are frequently absent from school despite being enrolled in schools [76]. Therefore, community outreach posts such as markets and community posts suggested in this study, stakeholder mobilisation, and other innovative ideas should be a policy priority to ensure that out-of-school girls are reached [77,78].

Our study makes a useful contribution, highlighting the perspectives of some key contextual stakeholders for HPV vaccine uptake, but further research is needed to generate more definitive evidence to inform decisions on which strategies to adopt in this setting. For instance, our NGT research could be repeated with broader groups of stakeholders in this setting (including programme managers and health administrators at facility and higher levels) to gain perspectives from a broader and more comprehensive group of stakeholders and see if there is consistency in the results with those from adolescents/parents/teachers/health workers in our study. Furthermore, future research could use the qualitative findings of our study to develop a quantitative survey and administer this on a larger sample of stakeholders, asking them to identify feasible strategies aligned to the contextual determinants in this setting. Future research could then use a longitudinal research design to test whether the identified tailored implementation strategies are feasible, acceptable, and increase HPV vaccine uptake (implementation effectiveness) in the Zambian context. 

### Strengths and Limitations

This study has strengths in that it builds on previous formative research and has engaged multiple stakeholders including out-of-school girls and creates a foundation for future research in Zambia and the region. The use of the NGT ensured that a rapid group consensus was achieved on common barriers. Additionally, the strategies selected were based on the context of the participants’ perception of feasibility and acceptability. Further, the ERIC taxonomy was used to code the identified strategies for easy replication and communication.

This study is not without limitations. Firstly, we documented barriers and strategies using the ranking method, rather than using frequency of use. However, the original NGT recommends ranking, which has also been used widely. Secondly, the sample size is small and non-random, which limits generalisability. Thirdly, the group consensus approach used may be skewed by dominant voices. However, the participants were aware of the HPV vaccine, and they all participated fully. Finally, though the study appropriately obtains the perspectives of key stakeholders involved in delivering or receiving the HPV vaccine, these participants selected strategies based on their experiences and perceptions rather than on evidence of whether their chosen strategies improve vaccine uptake. 

## 5. Conclusions

Implementation strategies should be tailored to the context where they are being implemented. In this study, key stakeholders were involved in identifying barriers important to them as well as selecting and tailoring the implementation strategies to these contextual barriers. The barriers identified are not unique to the Zambian setting. However, this process shows the important barriers to adolescent girls, parents, and teachers and healthcare workers in this set up. There was some between-group variation regarding what barriers were most important. 

The common barrier identified by all stakeholders was low awareness and knowledge about the HPV vaccine. Hence, from the perspective of our stakeholders, more educational strategies and other tailored strategies should be implemented. However, there is room for further research work to generate evidence for decision making on strategies to adopt in the Zambian context.

## Figures and Tables

**Table 1 vaccines-12-00542-t001:** Questions asked for identification of barriers.

No.	Stakeholder	Question
1	Adolescents	What are the barriers experienced by adolescent girls in receiving the HPV vaccine?
2	Parents	What barriers are experienced by parents to consent for their daughter’s HPV vaccination?
3	Healthcare workers and teachers	What are the perceived barriers for the implementation of the HPV vaccination program?

**Table 2 vaccines-12-00542-t002:** Characteristics of adolescent girls.

No.	Age	Goes to School?	School Grade	Received at Least One Dose of HPV Vaccine
1	16	Yes	11	Yes
2	15	Yes	10	No
3	15	Yes	10	No
4	15	Yes	9	Yes
5	15	Yes	9	No
6	16	Yes	11	No
7	15	Yes	10	No
8	15	Yes	9	Yes
9	15	Yes	9	No
10	15	Yes	9	No
11	18	No	NA	No
12	18	No	NA	Yes
13	18	No	NA	Yes

**Table 3 vaccines-12-00542-t003:** Identified barriers and tailored strategies by adolescent girls.

No.	Identified Barriers	Possible Implementation Strategies (Points)	Final Selected Implementation Strategies Tailored to Barriers—Voted on Based on Feasibility and Acceptability	Tailored Implementation Strategies Coded According to the ERIC Taxonomy [40]
1	Lack of knowledge and awareness about the HPV vaccine/program	- Use of TV, radio, flyers, newspapers, and social media to share information (65) - Conducting community educational meetings (63) - Educate adolescent girls in and out of school about HPV vaccine (58) - Conducting door-to-door campaigns on HPV vaccine (57) - Share information directed at adolescents (37)	- Use of TV, radio, flyers, newspapers, and social media to share information about HPV vaccine - Conducting community educational meetings - Educate adolescent girls in and out of school about HPV vaccine	- Use mass media - Conduct educational meetings - Involve consumers/patients and their families
2	Lack of school attendance	- Implement the free education policy (69) - Build more schools in communities (58) - Announce on TV, radio, and social media (52) - Increase school awareness and importance (39) - Create books on HPV vaccination (28) - Make more programs throughout the year (26)	- Implement the free education policy (69) - Build more schools in communities (58) - Announce on TV, radio, and social media (52)	- Purposely reexamine the implementation - Change service sites - Use mass media
3	Lack of parental consent to vaccinate daughters	- Educate parents on the advantages of the HPV vaccine (34) - Conduct door-to-door campaigns (33) - Parents should encourage children to get vaccinated (32) - Share information with parents (31)	- Educate parents on the advantages of the vaccine (34) - Conduct door-to-door campaigns on the HPV vaccine - Parents should encourage children to get vaccinated	- Conduct educational meetings - Increase demand - Involve patients/consumers and family members

**Table 4 vaccines-12-00542-t004:** Demographic characteristics of the parents.

No.	Age (Years)	Sex	Education Level	Employment	Child in School?	Child Vaccinated
1	28	Male	Primary	Yes	Yes	Yes
2	49	Male	Secondary	Yes	Yes	No
3	44	Female	Secondary	Yes	Yes	No
4	68	Male	Primary	No	Yes	Yes
5	60	Male	Secondary	No	Yes	No
6	23	Male	Secondary	No	Yes	No
7	34	Male	Tertiary	No	Yes	Yes
8	49	Female	Tertiary	Yes	Yes	Yes
9	25	Female	Secondary	No	Yes	No
10	52	Female	Secondary	No	No	No
11	54	Female	Primary	No	Yes	No
12	50	Female	Primary	No	Yes	No
13	59	Female	Secondary	No	No	Yes

**Table 5 vaccines-12-00542-t005:** Selecting and tailoring of strategies to identified barriers by parents.

No.	Identified Barriers	Possible Implementation Strategies (Points)	Selected Implementation Strategies Voted on Based on Feasibility and Acceptability	Tailored Implementation Strategies Coded According to the ERIC Taxonomy
1	Lack of knowledge and awareness about the HPV vaccine	- Coordinate health workers to work with the community (20) - Use of drama groups to raise awareness - Conduct door-to-door education on vaccines (27) - Offer incentives for vaccination program stakeholders (45) - Involve parents/girls (38) - Community education in places like schools, bars, and markets (36) - Distribute fliers and brochures which have information about vaccines to raise awareness (36)	- Offer incentive for vaccination program stakeholders (45) - Involve parents and girls (38) - Community education in places like schools, bars, and markets (36)	- Alter incentive for those involved in the vaccination - Involve patients/consumers and their families - Conduct educational meetings
2	Inappropriate venue set up for vaccination	- Secure a good place for vaccination which is near the target population (22) - Secure tents for vaccination venues for more privacy (14)	- Secure a good place for vaccination which is near the target population (22) - Secure tents for vaccination venues for more privacy (14)	- Change service site - Change physical structure and equipment
3	Myths and misinformation	- Educate people in the community about the HPV vaccine (41) - There should be more health talks in schools, markets, and churches (26) - Use of communication media like radio and television programs and drama to raise awareness (27) - Involve girls and parents (17)	- Educate people in the community about the HPV vaccine (41) - Use of communication media like radio and television programs and drama to raise awareness (27) - There should be more health talks in schools, markets, and churches (26)	- Conduct education meetings - Use of mass media - Change in service sites

**Table 6 vaccines-12-00542-t006:** Characteristics of teachers and healthcare workers.

No.	Age	Sex	Profession
1	48	Female	Nurse midwife
2	36	Female	Nurse
3	40	Female	Nurse midwife
4	42	Female	Nurse midwife
5	28	Male	Teacher
6	45	Female	Teacher
7	26	Male	Teacher
8	39	Male	Teacher
9	33	Female	Teacher
10	30	Female	Teacher

**Table 7 vaccines-12-00542-t007:** Selecting and tailoring of strategies by teachers and healthcare workers.

No.	Identified Barriers	Selected Implementation Strategies (Points)	Final Tailored Implementation Strategies Voted on Based on Feasibility and Acceptability	Tailored Implementation Strategies Coded According to the ERIC Taxonomy
1	Myths and misinformation in the community	- Adequate sensitization to the community, e.g., churches and schools using drama, music, and megaphones for community - Health education in churches, schools, and markets on HPV vaccine - Create one-on-one rapport with parents - Involve local celebrities and public figures - Give an incentive to those receiving the vaccine - Use local languages when sensitizing and simplifying reading materials on HPV vaccine - Conducting door-to-door sensitization with the help of community volunteers	- Use local languages when sensitizing and simplifying reading materials on the HPV vaccine (43) - Health education meetings in churches, markets, and schools on HPV vaccine (35) - Adequate sensitization to the community, e.g., schools and churches using drama, music, radio, and megaphones (33)	- Distribute educational materials on the HPV vaccine - Conduct educational meetings on the HPV vaccine - Use mass media
2	Lack of knowledge and poor sensitization among teachers and healthcare workers	- Conduct workshops for teachers and healthcare workers before the HPV vaccination exercise (40) - Provide more materials on the HPV vaccine (21) - Build teacher–healthcare worker rapport through communication on health matters (35) - Review what is discussed in meetings, e.g., quarterly (26) - Each head of department from the school should be part of the sensitization team (34) - Provide certificates of competence for those trained in giving HPV vaccinations (26)	- Conduct workshops for teachers and healthcare workers before the HPV vaccination exercise (40) - Build teacher–healthcare worker rapport through communication on health matters (35) - Each head of department from the school should be part of the sensitization team (34)	- Conduct ongoing training - Promote network weaving - Facilitation
3	Lack of information dissemination programs to the community	- The Ministry of Health should have strategies for disseminating information on radio and TV before a program starts (35) - Involve the school and clinic administration to come up with strategies to be used in the communities concerning HPV vaccination (26) - Provide handouts on HPV in various local languages (36) - Teachers should sensitize pupils to HPV for them to teach their parents (28) - Come up with short videos and movies about HPV to post on social media (34) - Make use of newspapers and magazines (26) - Engage influencers to educate people on HPV (39)	- Engage influencers to educate people on HPV vaccine (39) - Provide handouts on HPV in various local languages (36) - The Ministry of Health should have strategies for disseminating information on radio and TV before a program starts (35)	- Build a coalition - Distribute educational materials - Use mass media

## Data Availability

Data are available on special request from the corresponding author as this work is part of a PhD in progress.

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
