# Peer review of "Selecting and Tailoring Implementation Strategies to Improve Human Papillomavirus Vaccine Uptake in Zambia: A Nominal Group Technique Approach"

_vaccines, 2024, doi:10.3390/vaccines12050542_

Round 1
Reviewer 1 Report
Comments and Suggestions for Authors
This paper makes a useful contribution, but I think that it could be improved on a few points. First, there should be some comparison of the results for adolescents, parents, and teachers & healthcare workers. Although the questions asked of each group were not identical, they were similar enough so that some comparison should be possible. Were they consistent, or did there seem to be differences of emphasis or even conflicts among them? And if there were differences, could anything be done to reduce them? Second, there should be more detail about sampling. On p. 3, it speakds of homogeneous groups, but also of "diversity withing participants." On p. 4, it speaks of "purposive sampling," but doesn't say anything about the specific purposes that guided it. A third point is that this is an exploratory study--the sample is small and non-random, and although group discussion has some advantages, it also means that the responses may be strongly influenced by one or two persuasive members. So the obvious point is that this limitation should be acknowledged--that these results aren't definitive and that more research is needed. But it would also be helpful to say something about what kind of additional research would be most informative. For example, did the results suggest something about the kinds of questions that might be asked in studies applying similar methods, or in sample surveys?
Author Response
Dear Reviewer,
We are sincerely grateful for the time you invested in reviewing our manuscript. Your comments and suggestions have been very helpful in improving the quality of this manuscript. Kindly see below point by point responses to your comments.
First, there should be some comparison of the results for adolescents, parents, and teachers & healthcare workers. Although the questions asked of each group were not identical, they were similar enough so that some comparison should be possible. Were they consistent, or did there seem to be differences of emphasis or even conflicts among them? And if there were differences, could anything be done to reduce them?
Response: Thank you, there were some similarities, for example, both parents and adolescent girls ranked lack of knowledge about HPV and HPV vaccine as the most important barrier, while the healthcare workers and teachers ranked it as second. Similarly, myths and misinformation were ranked among the top three most important barriers by both teachers/healthcare workers group as well as the parents group. However, there were also differences in ranking which were unique to each group. Differences could be explained by the different levels of influence. For example, in many places, parental consent is required for adolescents to receive vaccinations. This adds an additional layer of decision-making and potential barriers if parents are hesitant or uninformed about the vaccine. I’m afraid that if we try to look for ways to reduce the differences, we will lose the context. Each of these stakeholders play different roles in the HPV vaccination cascade. Updated in paragraph one of the discussion page 3
Second, there should be more detail about sampling. On p. 3, it speaks of homogeneous groups, but also of "diversity within participants." On p. 4, it speaks of "purposive sampling," but doesn't say anything about the specific purposes that guided it.
Response: Thank you, homogenous in this context meant one type of participants only, e.g., adolescents only and not combined with parents. The diversity within the group part speaks to having in school and out of schoolgirls, as well as vaccinated and unvaccinated girls. The purpose was indicated as follows (lines 87 and 88): Participants were sampled purposively to select participants likely to give the most information about barriers to HPV vaccination in their context, and to select and tailor implementation strategies based on their feasibility and acceptability.
A third point is that this is an exploratory study--the sample is small and non-random, and although group discussion has some advantages, it also means that the responses may be strongly influenced by one or two persuasive members. So the obvious point is that this limitation should be acknowledged--that these results aren't definitive and that more research is needed. But it would also be helpful to say something about what kind of additional research would be most informative. For example, did the results suggest something about the kinds of questions that might be asked in studies applying similar methods, or in sample surveys
Response: Thank you, the limitation of the sample size has been acknowledged. Our findings are non-definitive and there is room for further work. We recommend future research to pilot test the different strategies including more stakeholder groups and see if there is consistency in results. Additionally, to pilot strategies using longitudinal research to test their effectiveness. Page 5 under conclusion
Reviewer 2 Report
Comments and Suggestions for Authors
1. Could each group consisting of only 10 to 13 people reflect the overall level of the research population? Please explain.
2. Could effective measures be proposed to improve HPV vaccination rates from the perspectives of Adolescent girls, parents, teachers and healthcare workers in the discussion section, respectively?
Comments on the Quality of English Language1. Line 20, The human papillomavirus (HPV) vaccine rather than "human papillomavirus (HPV)"?
2. Line 27, "The identified barriers included, low levels...", Remove the commas.
3. Line 31, "Tailored strategies included; conducting educational...", Remove semicolons and check punctuation mark throughout the entire manuscript.
4. Line 101, "We used the NGT a tool for consensus building in selecting and tailoring strategies", This sentence has a grammar error “NGT a tool” that needs to be rewritten, and the grammar of the entire manuscript needs to be checked.
5. Line 113, "the broader formative research (40)", What does this "40" in bracket represent, and the numbers that also appear in brackets in the following text? Although the numbers in brackets in the tables represent "points".
Author Response
Dear reviewer,
Please see attached.
Thank you for your review.

Reviewer 3 Report
Comments and Suggestions for Authors
This paper makes an important public health contribution. HPV is a serious public health concern, particularly in Zambia where this study was conducted. Discovering vaccination barriers as well as strategies for overcoming them are needed to address this public health challenge. The concept of tailored interventions has promise. The use of nominal group technique to generate barriers is appropriate and it produced useful information. As I indicate below, I have concerns about the ability of this approach to identify effective strategies for overcoming these barriers. Thus, there are major strengths to this article, although some concerns remain, most of which are minor.
Concerns
The paper needs to define “tailored.” It is used differently across the literature and we need to know what the authors mean by it. (minor concern)
Explain why you chose this particular setting within Zambia. (minor concern)
Were there any questions about what was meant by “barriers” in Step 1? (minor concern)
Why not have separate groups for vaccinated and unvaccinated participants? Please explain this choice. (minor concern)
What explains the difference in the definitions of barriers among the groups? How can these discrepancies be resolved? (minor concern)
I think the only relatively serious concern about this paper is the expectation that participants know the best strategies that can be used to promote vaccination. In my opinion, this is a theoretical and empirical question not one for lay audiences. People tend to be poor explainers and predictors of their own behaviors so expecting them to know how best to promote vaccination is, I believe, severely limited in value. For example, if by education you mean information, this is often an ineffective strategy for changing adolescent behavior. You note this in your discussion – I believe you should tie it to the inability of participants to identify evidence-based practices. Similarly, we need to know about the content of the magazine intervention. Narrative interventions have proven effective in increasing uptake – were there narratives in the magazine or was it more didactic?
Round 2
Reviewer 1 Report
Comments and Suggestions for Authors
This revision is a reasonable response to my comments, although I'd like to see a bit more on the last point--trying to be more specific about what kind of additional research is needed. There's also an important typo on line 317: it says "This study is without limitations" when presumably they mean to say "this study is not without limitations"
Comments on the Quality of English LanguageNo issues
Author Response
Dear reviewer,
Thank you so much for your thoughtful review.
We have updated the manuscript with specifics on future research. See the text below which has been added to the manuscript discussion section.
The typo on limitations has been corrected.
Response: Our study makes a useful contribution, highlighting the perspectives of some key contextual stakeholders for HPV vaccine uptake, but further research is needed to generate more definitive evidence to inform decisions on which strategies to adopt in this setting. For instance, our NGT research could be repeated with broader groups of stakeholders in this setting (including programme managers and health administrators at facility and higher levels) to gain perspectives from a broader and more comprehensive group of stakeholders and see if there is consistency in results with those from adolescents/parents/teachers/health workers in our study. Furthermore, future research could use the qualitative findings of our study to develop a quantitative survey and administer this on a larger sample of stakeholders asking them to identify feasible strategies aligned to the contextual determinants in this setting. Future research could then use a longitudinal research design to test whether the identified tailored implementation strategies are feasible, acceptable, and increase HPV vaccine uptake (implementation effectiveness) in the Zambian context.
Reviewer 2 Report
Comments and Suggestions for Authors
The page numbers in the entire text are incorrect.
Comments on the Quality of English Language 1. Line 63, "A description of the Implementation of the HPV vaccination" , “The first letter of “Implementation”should be lowercase, and check the entire manuscript.2. Line 317, "This study is without limitations, firstly we...". This sentence is expressed incorrectly.
Author Response
Dear reviewer,
Thank you so much for the feedback. Here are the point-by-point responses.
Overall, the manuscript has been improved with primary focus on the discussion, strengths and limitations and the conclusion with support from co-authors.
- Line 63, "A description of the Implementation of the HPV vaccination" , “The first letter of “Implementation” should be lowercase and check the entire manuscript.
Response: This has been done and other areas needing correction have been attended to with support from co-authors.
2. Line 317, "This study is without limitations, firstly we...". This sentence is expressed incorrectly.
Response: The correction has been made
3. Page numbers
Response: Attempts have been made to do this; however, the template is tagged by the editorial office hence we have not been successful-the confusion came about as we had to make section breaks to align some tables in landscape. However, we have included this challenge in our response to the editor for correction of the page numbers during typesetting of the manuscript.
Thank you.